# The concordance of signals based on irregular incremental lines in the human tooth cementum with documented pregnancies: Results from a systematic approach

**Gabriela Mani-Caplazi**[1], **Werner Vach** [1,2]*, **Ursula Wittwer-Backofen**[3], **Gerhard Hotz**[1,4]

**1** Integrative Prehistory and Archaeological Science, University of Basel, Basel, Switzerland, **2** Basel Academy for Quality and Research in Medicine, Basel, Switzerland, **3** Biological Anthropology, University of Freiburg, Freiburg, Germany, **4** Natural History Museum of Basel, Anthropological Collection, Basel, Switzerland

* werner.vach@unibas.ch

**Data Availability Statement:** All relevant data are within the manuscript and its Supporting Information files.

## Abstract

### Background and objective

There is evidence from previous studies that pregnancies and diseases are recorded in the tooth cementum. This study aims to assess the degree of concordance between signals based on irregular incremental lines (ILs) and reported pregnancies.

### Material and methods

23 recent and 24 archaeological human teeth with known birth history were included in this investigation. 129 histological sections of tooth roots were assessed for irregularities in appearance and width using a standardized protocol. Similarity of observed irregularities at the section level allowed us to define signals at the tooth level. The sensitivity of signals to detect pregnancies was determined and related to the signal prevalence.

### Results

Pregnancy signals were frequently visually observed. However, applying a standardized process we could only reach signal sensitivities to identify pregnancies up to 20 percentage points above chance level.

### Conclusions

Based on a standardized and reproducible method it could be confirmed that some pregnancies leave visible signals in the tooth cementum. The results show the potential of the tooth cementum to support reconstruction of life courses in paleopathology. However, it seems that not all pregnancies affect the cementogenesis in such a way that irregular ILs are identifiable. Further research is needed to better understand which type of pregnancies and other conditions are recorded in the tooth cementum.

**Funding:** The author(s) received no specific funding for this work.

**Competing interests:** The authors have declared that no competing interests exist.

## Introduction

Estimating fertility is a major concern in anthropological science. Having access to the pregnancy history of an individual provides the means to estimate age at pregnancy, number of pregnancies and pregnancy intervals. Several studies in animals [1–4] and in humans [5–9] were able to identify reproduction based on permanent records in the tooth cementum. (An overview is presented in [10]). Tooth cementum is deposited lifelong, supporting the anchoring of the tooth in the alveolar bone, and shows a layered structure with incremental lines (ILs) [11]. One IL is assumed to represent one year [12, 13]. Cementum is rarely remodelled [14] and preserves the structure including irregularities which may point to events in life history [15]. These properties allow identified irregularities to be dated based on the tooth cementum annulation (TCA) [16]. This is done by counting ILs in the acellular extrinsic fiber cementum, the cementum used for TCA [17], and adding it to the mean tooth eruption age per tooth type [18]. However, several questions, in particular in humans, are still open as to how consistent and precise pregnancies can be identified.

In general, paleopathological examinations aim at reconstructing health conditions of ancient populations based on identifiable alterations in the human remains. Pregnancy is a physiological process however is a demanding condition with changes in the maternal metabolism [19, 20]. There is an increased need for calcium and a vulnerability period for maternal bones which can even lead to osteoporosis [21]. Due to the structural similarity between bone and tooth cementum [14] it is assumed that pregnancies can alter the tooth cementum growth [22] and may leave permanent markers in the tooth cementum. It has been observed that renal disorders and skeletal trauma affect the tooth cementum [5]. Tooth cementum, with its layered structure allows such events to be dated and has the potential to be a comprehensive repository for paleopathologists to retrieve information in reconstructing the life course and health condition of ancient populations.

Animal reproduction could be identified in some studies with a precision higher than 70% [2, 3]. However, in humans the identification of pregnancies based on ILs in the tooth cementum is more challenging considering the higher number and thinner ILs along with the less clear life cycle. In previous studies, such markers in the tooth cementum were called stress marker [6, 23]. A systematic assessment of the correlation of such stress markers in the tooth cementum with pregnancies has not yet been done in humans and it is unclear if all or just certain pregnancies are visible in the tooth cementum. Pregnant women comprise a vulnerable population and it could be that in particular more demanding pregnancies lead to stress markers in the tooth cementum.

A systematic assessment of irregularities related to ILs in the tooth cementum of the teeth has been previously conducted blinded for any data on the birth history [24]. Irregular ILs were reproducible to a moderate degree across sections, however showed a plausible pattern very likely pointing to biological influences.

In the present study we aim to investigate the concordance between pregnancies and irregular ILs in these teeth building on the identified irregular ILs in the above mentioned first assessment. This requires two basic steps. The first step is to obtain signals defined at the tooth level. This requires a process to define which irregular ILs across sections may correspond to a common signal at the tooth level. In the second step, for the comparison of such signals with the age at pregnancy, we have to assign an age to each signal based on the number of ILs, an estimate of the eruption age and the age at death/extraction. These two steps follow a structured approach, following a clear protocol taking the visual similarity of ILs across sections–into account in order to define signals. This visual inspection should help to identify relevant irregularities that are present with a certain consistency across sections.

The objective of the study is to assess the degree of concordance between signals based on irregularities in ILs and reported pregnancies in recent and archaeological samples. The focus will be on the ability to detect pregnancies in a reliable manner and hence to reconstruct the birth history.

## Material

Our investigation is based on 47 teeth from different individuals and two different samples (recent and archaeological). A concordance of identified signals in the tooth cementum and known pregnancies have so far been reported from recent human teeth [5, 6] and we wanted to assess if pregnancy signals are also identifiable in archaeological teeth which can be exposed to post mortem taphonomic processes. Also the differences in life conditions are an opportunity to investigate with these two populations. The teeth from the recent population originate from a dental clinic in Germany. Date of birth of the women and date of tooth extraction were taken from the medical records. The women were directly asked about the number of births (including stillbirths and abortions) and the corresponding dates and verbal consent was provided to use this information for research purposes [6]. The teeth of the archaeological population were sampled from the Basel-Spitalfriedhof collection from the nineteenth century stored at the Natural History Museum Basel, Switzerland [25]. Date of birth, date of death and the cause of death of the women were obtained from the medical records and number and date of births (and stillbirths) were determined based on various types of historical sources. In addition, each birth was classified as legitimate or illegitimate depending on whether the woman was married or not at the time of birth. Details of this process are described in S1 Text. In both populations also the gestational age was recorded in case of stillbirths and abortions.

Basic characteristics of the teeth/individuals and the pregnancies are shown in Table 1. Overall, 80 pregnancies were reported with a date of birth, stillbirth or abortion. Two additional pregnancies in women from the recent population were reported without date.

The teeth had been embedded in synthetic resin and up to five consecutive cross sections per tooth from the apical part of the middle third of the tooth root towards the crown were prepared applying the technique described in [17]. Then sections (typically three) with the best IL quality were selected according to criteria described in the supplementary information of [24]. These selected sections were then examined by transmission light microscopy in ×25 to ×400 magnification and areas with good IL presentation were scanned using a digital camera.

The sections from each tooth were systematically investigated for the presence of irregular ILs, i.e., potential signals. In each section ILs were identified and numbered and for each IL the width and the presence of Appearances (i.e., visually irregular appearing ILs, darker or brighter than the neighbouring ILs) were assessed from three different rows of measurements. The width measurements were used to determine relevant local deviation from the normal width. These were defined as local peaks, i.e., single ILs with a width distinctly higher than the width of neighbouring ILs within each row. The peaks were classified as 1SD, 2SD and 3SD peaks, reflecting whether the width was increased by at least 1, 2, or 3 standard deviations compared to the local average. The Appearances were classified as absent, weak or strong, defining an Appearance Index with the values 0, 1, and 2. At the level of each section local peaks were defined in an analogous manner based on the mean width, and an Appearance Sum Index was defined as the sum over the three row-specific values of the Appearance Index. In addition, the quality of each IL with respect to the suitability to measure the width was classified as low, medium, or high. Details of this procedure are described in [24].

As an additional source of information we used the publication of [18] to obtain estimates of gender- and tooth-specific average tooth eruption ages.

**Table 1. Basic characteristics of individuals/teeth and the pregnancies included in this study.**

| | recent | archaeological |
|---|---|---|
| Number of individuals/teeth | 23 | 24 |
| Age at extraction/death | | |
| mean(SD) | 51.3 (14.4) | 45.6 (15.8) |
| min-max | 18–78 | 25–74 |
| Tooth code (FDI numbering system) | | |
| 12/22 | 3 | 1 |
| 13/23 | 2 | 1 |
| 14/24 | 1 | 0 |
| 15/25 | 1 | 0 |
| 31/41 | 6 | 1 |
| 32/42 | 6 | 3 |
| 33/43 | 1 | 13 |
| 34/44 | 0 | 1 |
| 35/45 | 1 | 2 |
| Number of sections analysed | | |
| 2 | 9 | 3 |
| 3 | 14 | 21 |
| Number of pregnancies per mother (births and abortions) | | |
| 0 | 1 | 8 |
| 1 | 7 | 7 |
| 2 | 11 | 6 |
| 3 | 2 | 1 |
| ≥4 | 2 | 2 |
| Number of abortions per mother | | |
| 0 | 22 | 21 |
| ≥1 | 1 | 3 |
| Number of pregnancies | 42 | 38 |
| Age at pregnancy: | | |
| mean (SD) | 23.6 (5.5) | 28.1(5.3) |
| min-max | 14–42 | 21–42 |
| Legal status of the child | | |
| legitimate | - | 17 |
| illegitimate | - | 21 |

## Methods

### The process

The starting point is to translate previously identified irregular ILs from [24] in terms of width and appearance into signals. To support this the identified irregular ILs were displayed in an IL width growth curve, an example of which is shown in S2 Text. These curves facilitate the identification of potential signals across sections. The identified signals are then compared with the original images of the tooth sections. This renewed visual inspection–taking into account the visual similarity of signals across sections–leads to a refinement of the signals. This visual inspection is relevant considering that signals are not present in the identical ILs across sections. Details are described in S2 Text. The output from this process is exemplified in Table 2 for one tooth. For each signal identified, the ILs involved in each section and two indices are given. The first index describes the signal intensity at each section and the second the

**Table 2.  Final output of the signal identification based on the incremental line growth curves (example: Z_131) and the microscopic section images.**

| Tooth/Section number | Image number | Signal number | IL number | Signal intensity per section | Signal matching across sections |
|---|---|---|---|---|---|
| Z_131 | R_8251953 | 1a | 2 | + | + |
|  |  | 1b | 5 | 0 | + |
| Section 1 |  |  | 6 |  |  |
|  |  | 2 | 18 | + | ++ |
|  |  |  | 19 |  |  |
| Z_131 | R_95413208 | 1a | none | n/a | n/a |
| Section 2 |  | 1b | 4 | + | + |
|  |  | 2 | 18 | ++ | ++ |
| Z_1314 | R_14498901 | 1a | 2 | + | + |
|  |  |  | 3 |  |  |
| Section 3 |  | 1b | 4 | ++ | + |
|  |  | 2 | 19 | + | ++ |
|  |  |  | 20 |  |  |

matching of the signals across sections for the tooth. Both indices are scored at three levels (0 = low, + = medium, ++ = high) and take both the visual appearance as well as the irregularity in IL width into account.

### Three cases illustrating the data analytic challenge

Three cases presented in S3 Text illustrate the challenge we are confronted with in analysing the available data. On one hand, the exact dates of the pregnancy are known for each woman, hence the age at pregnancy is available. On the other hand, we have potential signals associated with irregular ILs and we have to translate them into ages. We present cases for which the identified signals were matched visually (not following any specifications) with documented pregnancies by assuming one IL per year since the average eruption age of the tooth [18]. However, the cases illustrate that this has limitations and the allocation of signals to pregnancies in this way is very subjective. Additionally, the number of ILs do not match exactly to the age range between assumed eruption age and date at death/extraction. Consequently, we need to allow signals to cover a time range of more than one year in order to catch all pregnancies. Moreover, the cases illustrate that the potential signals may vary highly in their intensity, and it is unclear, which intensity is required to regard them as convincing signals, in particular as we usually have more signals than pregnancies. So, we need to define thresholds for signal intensity. All these issues underline the need to have some objective methodology to evaluate the concordance between pregnancies and signals. These cases also illustrate that some of the potential signals may be due to causes other than pregnancies, and that the agreement between pregnancies and signals may vary from subject to subject.

### Overview about the further analytic strategy

The basic aim of our investigation is to check whether we can detect pregnancies based on identified irregularities in the tooth cementum. This requires to transform such irregularities into potential signals at the level of each tooth/individual and to assign an age (or age range) to each signal, which can be compared with the age at pregnancy. A first list of potential signals at the tooth level has been created by the process described above (3.1) with a range of ILs involved in each section. Three further steps are necessary: 1) The signals vary in intensity in relation to irregular appearance and width (local peaks). Only signals of a certain degree of

intensity or a certain type may hint to pregnancies. Hence, we have to define different subsets based on specific characteristics, which we call "source signals". 2) To each IL we assign an age range based on the IL number, on an estimate of the eruption age [18] and on the extraction/ death date. 3) The IL ranges of each source signal are translated into age ranges at each section and the age ranges from the different sections are merged into one age range at the tooth level.

Below we describe the details of each step. Since we consider in each step different variants for the corresponding signals, we arrive at the end at a large variety of derived signal variants, and we will evaluate each derived signal variant separately. The following will be considered: First we check whether we can find on average more signals close to a pregnancy than in some distance to a pregnancy. Second, we will consider how often on average a pregnancy is detected by the signals according to a specific variant, i.e. how often the age of a pregnancy is covered by the age range of some signal. In addition, the second step is used to study whether certain characteristic of the pregnancy, of the mother, or of the tooth have an influence on the probability of a pregnancy in being detected.

In addition, we consider an approach avoiding to assign ages to signals: A the tooth level we compare the overall signal intensity (i.e. the number of signals) with the number of pregnancies.

## Defining variants of source signals

The identification of the signals included a grading of the intensity of the signals (S2 Text, step 5 and 6). This leads to different variants depending on the intensity. In addition, the identification was based on the presence of Appearances or local peaks. Hence it is of interest to distinguish between signals associated with Appearances and signals associated with local peaks. Consequently, we consider twelve source signal variants shown in Table 3.

## Assigning age ranges to IL numbers

Each IL number represents a certain period in the life of an individual. The use of ILs in age determination suggests that each IL roughly represents one year. To link an IL number to a certain age, we have two benchmarks: An estimated tooth eruption age, which we can relate to the first observed IL and the age of extraction/death, which we can relate to the last observed IL. To estimate the age corresponding to each IL we have to add the estimated tooth eruption age [18] to the number of the respective IL. However, since the number of ILs identified need

**Table 3. The source signal variants.**

| Signal variant | |
|---|---|
| S1) | All signals |
| S2) | All signals with across section index = ++ or + |
| S3) | All signals with across section index = ++ |
| S4) | All signals with index of signal intensity = ++ for at least two section or all sections |
| S5) | All signals with index of signal intensity = ++ or + for all sections |
| S6) | All signals with index of signal intensity = ++ or + in at least two sections or all sections |
| S7) | All signals with an Appearance Sum Index ≥5 for one IL number for at least two sections or all sections |
| S8) | All signals with an Appearance Sum Index ≥4 for one IL number for at least two sections or all sections |
| S9) | All signals with an Appearance Sum Index ≥3 for one IL number for at least two sections or all sections |
| S10) | All signals with a 1SD peak with respect to the mean width for one IL number of at least two sections or all sections |
| S11) | All signals with 1SD peaks in at least two rows for one IL number for at least two sections or all sections |
| S12) | All signals with 1SD peaks of non-low quality in at least two rows for one IL number for at least two sections or all sections |

not to coincide with the number of years between the two benchmarks, the assignment of age ranges is not completely straightforward. We consider four variants to assign non-overlapping age ranges to IL numbers. They are all based on:

- Assigning the average eruption age to the middle of the age range of the first observed IL

- Assigning the extraction/death age to the end of the age range of the last observed IL

However, the assignment of age ranges can vary with respect to the ILs between the two benchmarks. We consider the following variants:

L1) Assuming a length of one year for each IL starting from the first IL

L2) Assuming a length of one year for each IL starting from the last IL

L3) Using L1) up to the middle IL number and L2) for the remaining IL numbers

L4) Choosing the length of the age range of each IL such that the overall actual age range of a
   section matches the difference between the two benchmarks.

Whenever variant L3 results in an overlap (because there is more than 1 IL per year), L3 is replaced by L4, i.e. the age ranges are shrunk until there is no longer any overlap. Note that all these computations regard any age as an exact age in days.

However, the uncertainty mentioned above also suggests allowing wider age ranges for each IL in order to increase the likelihood that a signal at an IL covers the age of a corresponding pregnancy. This widening is thought to be inevitable considering that the estimated age based on the numbers of ILs and the actual age do not fully match in most cases. Therefore, we consider two strategies to widen the age range. First, we consider widening the age range symmetrically by a factor F to take a general uncertainty into account. We consider the choices F = 1, 2, 3, 4 and 5. We perform this widening symmetrically in both directions until the desired width is reached. Second, we can take into account the difference $\Delta$ between the number of years between the two benchmarks and the number of ILs identified. The larger the difference the larger may be the need to widen the age range of a signal in order to allow a pregnancy to be detected. The difference itself indicates the maximal shift we may have to take into account for a single signal. If we further assume that the "missing" ILs are roughly uniformly distributed across the ILs observed, we can also come to a more precise suggestion for the widening based on the relative position $r$ of the IL among all ILs of a section. The definition of $r$ depends on the choice of L. When using L1 the first IL has the relative position 0.0 and the last one the relative position 1.0, and the necessary amount of widening starts at 0 and ends with the value $\Delta$. In the case of L2, the definition of $r$ is reversed. If L3 or L4 are used, we switch in the middle of the IL range from the first to the second definition. However, we cannot be sure that these considerations are exactly valid, hence we also consider widening all age ranges by the maximal amount over all values of $r$.

We thus consider three variants of an additional widening on top of the application of the factor F:

W0) No additional widening

W1) Widening the age range of an IL by $r \times \Delta$

W2) Widening the age range of each IL by $\Delta$ (L1 and L2) or $\Delta/2$ (L3 and L4) independent of $r$

The widening is performed to the right for L1 always and for L3 in the lower half, to the left for L2 always and for L3 in the upper half, and by a symmetric widening in both directions for L4.

## Generation of derived signals from the source signals

The process described above characterizes each original signal (and thus each source signal) by a range of ILs in each section. We can translate these IL number ranges to age ranges by taking the union of the IL number specific age ranges. These section-specific age ranges must still be merged into one tooth-specific age range. We consider here three variants:

D1) Using the union of all date ranges (i.e. from the minimal lower bound to the maximal upper bound)

D2) Using the average lower bound and the average upper bound

D3) In the case that a signal is present in all three sections, we use the range from the minimal date present in at least two sections up to the maximal date present in two sections. If such a date does not exist or if the signal is present in less than three sections, variant D1) is used.

## Age of pregnancy and risk times

Each documented pregnancy, including abortions and stillbirths, is taken into consideration. For the analyses each pregnancy is reduced to a single age, defined as the birth date minus six months–reflecting the increasing change in the maternal metabolism during pregnancy and in particular the rising calcium demand related to mineralization of the foetal skeleton [26]. For stillbirths and abortions information on the gestational age is used to approximate this time point. For nine pregnancies in the recent sample and one pregnancy in the archaeological sample only the birth year, but not the exact date is given. Here the 1st of July is assumed.

The possibility of pregnancy is considered for each individual starting with the 13th birthday and ending with the 45th birthday or the date of extraction/death, whichever comes first.

## Distance analysis

For each signal variant we consider the presence of signals as a function of the distance from a pregnancy. So, we consider the simple function.

$d \mapsto$ frequency to find a signal when looking d days apart from each pregnancy.

We will consider both positive and negative values of d. We determine the frequency for each value of d in the range from -15 to +15 years with a spacing of 30 days in between. We then smooth these values by applying a lowess smoother [27] and show the resulting values in the range from -120 months to 120 months.

## Sensitivity vs. prevalence analysis

For each signal variant we consider two quantities:

- Sensitivity: The fraction of pregnancies with an age covered by an age range of a signal.

- Signal prevalence: The sum of all age ranges of the signals (corrected for overlap) divided by the overall risk time, i.e. the sum of the individual risk times

In general, sensitivity increases with signal prevalence: The longer the overall time span of the age ranges of all signals according to one variant, the more pregnancy ages will be covered by these signals. Therefore, our interest is in signal variants with a high sensitivity and low signal prevalence.

## Comparison with chance level

For both analytic approaches we perform a comparison with the chance level. This is approached by generating for each individual random pregnancies uniformly distributed over the risk period. The number of generated pregnancies in each individual is equal to the observed number of pregnancies. We repeat this 200 times and compute the mean and the upper 95% percentile of the sensitivity or the frequency at each distance d, respectively. We refer to the values as the expectation at chance level and as the 95% limit at chance level.

## The influence of characteristics of the pregnancy, of the mother, or of the tooth on the detection of a pregnancy

The probability of a pregnancy to be detected by irregularities of the ILs may depend on characteristics of the pregnancy, the mother or the tooth. Hence, we also investigate the hit frequency of each pregnancy, i.e. the frequency of each pregnancy to be covered by the age range of a signal when considering all signal variants. As characteristic of the pregnancy, we consider the age at the pregnancy age and the birth order, i.e. the number of previous pregnancies including the current one. In addition, we also consider among the individuals in the archaeological sample the legal status of the child at birth (legitimate or illegitimate). As characteristic of the mother, we consider the overall number of pregnancies reported and the distinction between the recent and the archaeological samples. As characteristic at the tooth level we consider the placement in the upper or lower jaw and the tooth type defined as the second digit of the FDI tooth numbering system, ranging from 1 for the central incisor to 8 for the 3rd molar. Finally, we consider a methodologically motivated characteristic: The relative age of the pregnancy within the risk period, ranging from 0 at the start to 100 at the end of the risk period. Here we expect that pregnancies close to the border are better to detect due to less difficulties in placing the ILs correctly in the life span.

## Association of signal intensity with number of pregnancies

The association between pregnancies and source signals can be also investigated without assigning an age to each signal: The overall signal intensity in a tooth can be correlated with the number of pregnancies. Accordingly, we consider for each of the 12 different variants in defining source signals the number of detected signals as a measure of signal intensity.

We restrict the computation of the number of signals to the signals in the in the risk period of each individual. We use here the variant L3 to assign an age to each IL, which is likely to be most accurate at an age of 13 and 45, respectively. We do not perform any widening and use variant D2 to determine an age range and then use the middle of the age range as the age of the signal.

To depict the association, we model each measure of signal intensity as a linear function of the number of pregnancies and the average number of ILs available. The regression coefficient of the number of pregnancies then describes the average increase in number of signals with each pregnancy.

The two pregnancies with missing information of the age of birth are included in computing the number of pregnancies.

## Performance of the different variants in the construction steps of the derived signals

To inform future investigation about the optimal way to derive signals based on the source signals, a systematic comparison of the variants considered is desirable. The methodology for

such a comparison is presented and applied in (S4 Text). Comparisons of the different source signal variants will already be performed as part of the sensitivity-prevalence analyses.

## Statistical significance level

The statistical significance level is set to 5%. All statistical computations have been performed with Stata 16.1.

## Ethical considerations and dual publication

With respect to the recent sample, the participants provided informed verbal consent but the consent was not documented. There was no separate requirement for approval by an ethics committee. With respect to the archaeological sample from 19th century Switzerland, the use of the data for research purposes has been accepted by the data protection officer of the city of Basel. The data protection officer confirmed that—according to the Swiss law—working with such data does not require additional specific permission, institutional approval by an ethics committee, or an IRB.

The original version of the manuscript is part of the first author's PhD thesis, which has been accepted by the University of Basel. The thesis will be made publicly available in March 2023.

# Results

## Number of source signals

For the 12 variants in defining source signals we observe the frequencies shown in Table 4. Compared to the overall number of 80 pregnancies the variant S1 (all possible signals) defines about twice as many signals. Four variants lead to a number of signals comparable with the number of pregnancies (S2, S5, S6, S9), whereas all other variants lead to lower numbers.

## Relation between number of ILs and age at extraction/death

Fig 1 depicts the relation between the number of ILs observed and the age at extraction/death. For each section the number of ILs is transformed into an age by adding the average eruption age [18]. We observe a substantial variation in the number of ILs per section for some

**Table 4. The frequency of the source signals in the range from the 13th to the 45th birthday.**

| Variant | Frequency |
| --- | --- |
| S1 | 163 |
| S2 | 96 |
| S3 | 30 |
| S4 | 24 |
| S5 | 83 |
| S6 | 99 |
| S7 | 17 |
| S8 | 37 |
| S9 | 82 |
| S10 | 50 |
| S11 | 35 |
| S12 | 22 |

Note: For the source signals, all signals are considered for which the middle of the age range of the derived signal according to L3, D2, F1 and W0 is between the 13th and the 45th birthday.

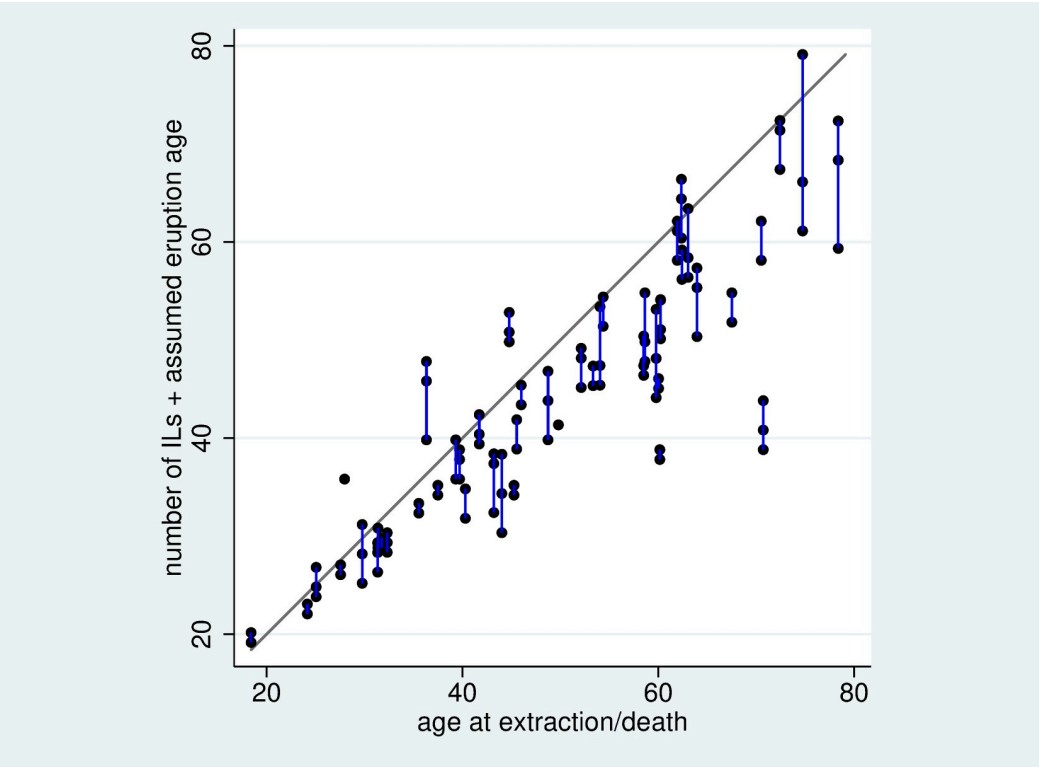

**Fig 1. Scatterplot of the reconstructed age according to the number of ILs and the assumed age at eruption based on [17] versus the true age at extraction/death.** As the number of ILs varies from section to section, each individual is represented by 2 or 3 points connected by a blue line. The grey line indicated equality of x and y to determine the age.

individuals. In general, there is a tendency to an underestimation of the true age when using the number of ILs as a surrogate, with a tendency towards an increasing difference with increasing age. This may suggest that in some individuals we miss some of the ILs. However, in some individuals we observed more ILs than expected given the true age.

## Analysis of signals

Fig 2 shows results of the distance analysis for one choice of L, F, W, and D for all S-variants of the source signal according to Table 4. We observe for all variants a distinct peak in the number of signals over the whole range of distances considered. However, we observe such a peak also at chance level. It is hence essential to compare the peak observed with the expectation and the upper 95% limit at chance level. For the S-variants S1, S2, S5, S6, S8, S9, S10, and S12 we observe a peak distinctly above the 95% limit. For the variants S3, S4 and S11 the peak does not exceed the 95% limit.

Fig 4 gives an overview about the results of all sensitivity-prevalence analyses. Only few S-variants reach a sensitivity of 20 percentage points above the expectation at chance level, and this only for few choices of the other variants L, F, W, and D. On average, only the variants S2, S8 and S9 reach an increase by 10 percentage points. However, S8 reaches this at a much lower prevalence.

## Association of pregnancy, mother or tooth characteristics with hit frequencies

In Fig 5 we can observe an increase of the hit frequency with birth order, overall number of pregnancies, location of the tooth in the upper jaw and the tooth type. However, these associations do not reach statistical significance. With respect to the relative age of the pregnancy we

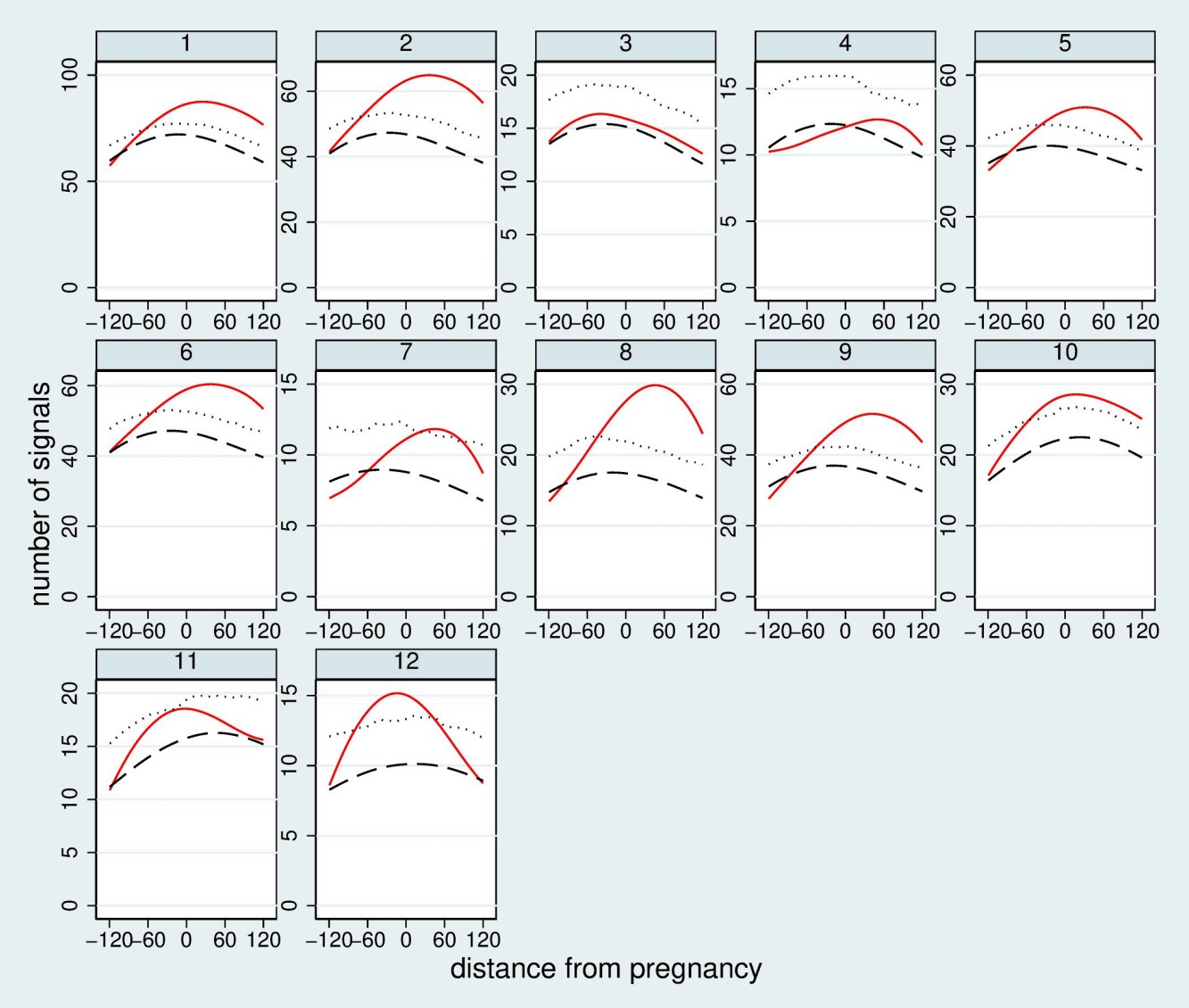

**Fig 2. Results of the distance analysis for L1, F4, W0, and D1 and all S-variants of the source signal according to Table 4.** The red line indicates the observed number of signals with a certain distance (in months) from the pregnancy. The black dashed line indicates the expectation at chance level. The dotted line indicates the upper 95% limit at chance level. Fig 3 shows results of the sensitivity-prevalence analysis for two choices of L and one of F, W and D, respectively for all S-variants of the source signal. A wide range of prevalence values is observed reflecting the differences in the number of source signals per variant. For L1 most variants imply a sensitivity above the expectation at chance level and even above the 95% limit at chance level. We reach maximally an increase in sensitivity about 20 percentage points compared to the expectation at chance level. When considering variant L4, only few variants of the source signal show a sensitivity distinctly above chance level.

observe the expected U-shape, i.e. better detection when a pregnancy is closer to the eruption or extraction/death date. Within the archaeological population hit frequencies we could observe no difference on average between legitimate and illegitimate births, however; the pregnancies with highest detection frequency were always illegitimate births.

## Association between number of pregnancies and signal intensity

Table 5 depicts the association of the signal intensity measures with the number of pregnancies. We observe for eleven of the twelve variants a positive association. They are most pronounced for

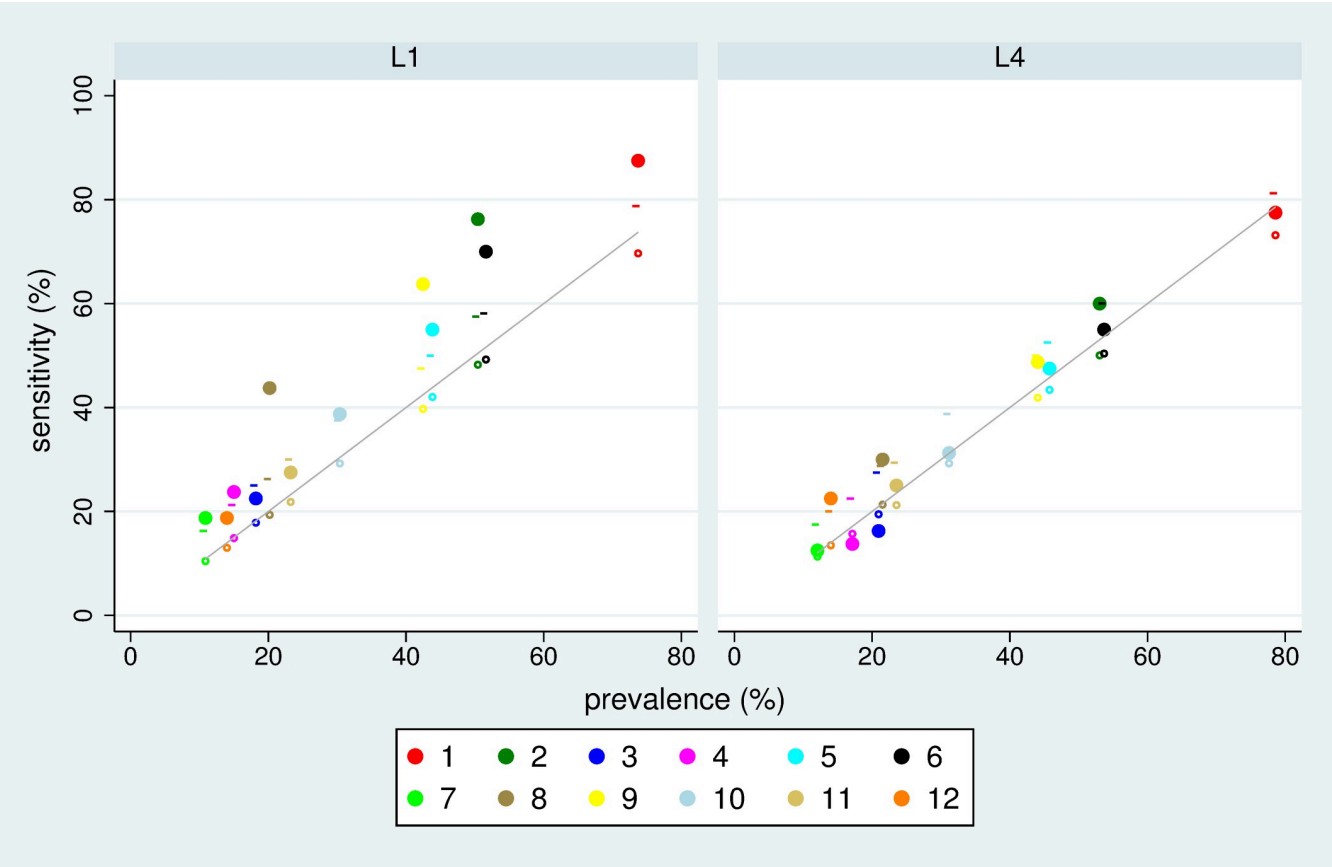

**Fig 3. Scatterplots of sensitivity versus prevalence for the 12 S-variants in defining the source signal and the choice F4, W0, D1, and L1 or L4.** The small circles indicate the expected sensitivity at chance level, and the small vertical lines the upper 95% limit at chance level. The grey line indicates equality of sensitivity with prevalence.

variant S2 suggesting the occurrence of this signal type at each 5th pregnancy, and for variant S8, suggesting the occurrence at each 7th pregnancy. However, none of these associations reaches statistical significance. Fig 6 illustrates some of most distinct observed associations.

### Performance of the different variants in the construction steps of the derived signals

In S4 Text we present a systematic comparison of the performance of the different variants with respect to defining signals with a good sensitivity to prevalence relation. In summary, it could be shown that some widening of the age range is essential in order to obtain a good performance, but the optimal degree remains uncertain. The variants L1 and L2 and D1 seem to be superior to the other choices, but the differences are small. The choice of the source signal variant seems to be the most relevant aspect.

## Discussion

### Summary of results: Detection of pregnancies by irregularities of ILs in the tooth cementum

Our investigation confirms previous reports about an association between pregnancies and irregular ILs in the tooth cementum. Distance analyses indicated that we can find irregularities

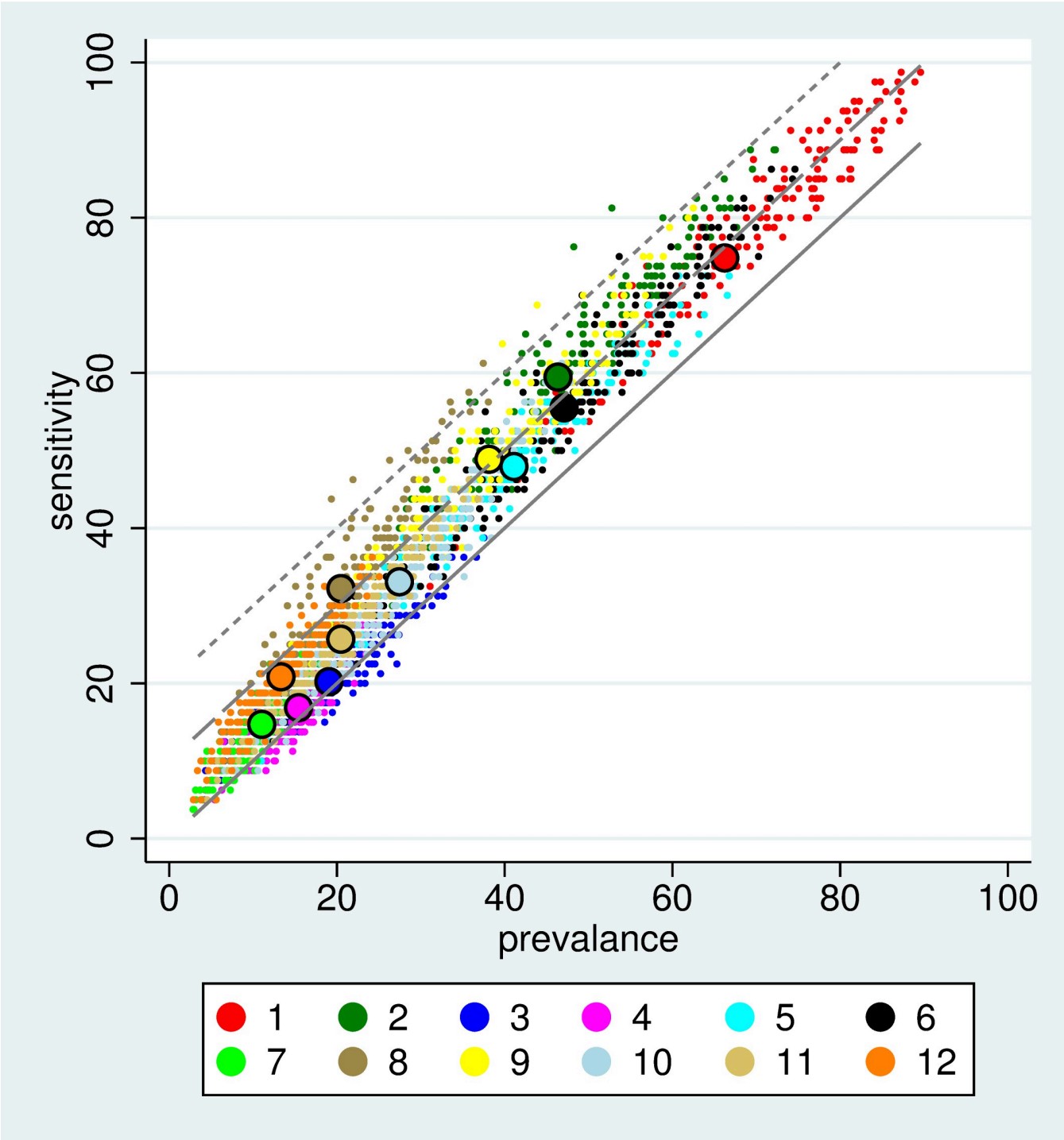

**Fig 4. Overview of all sensitivity vs. prevalence analyses.** Each small dot corresponds to a sensitivity/prevalence pair observed for one combination of the variants of L, F, W, D, and S. The colours correspond to the different S-variants in defining the source signal. Each big dot corresponds to the mean prevalence and mean sensitivity for one S-variant over all other variants. The three parallel grey lines refer to a difference between sensitivity and prevalence of 20 (dotted), 10 (dashed) and 0 (solid) percentage points.

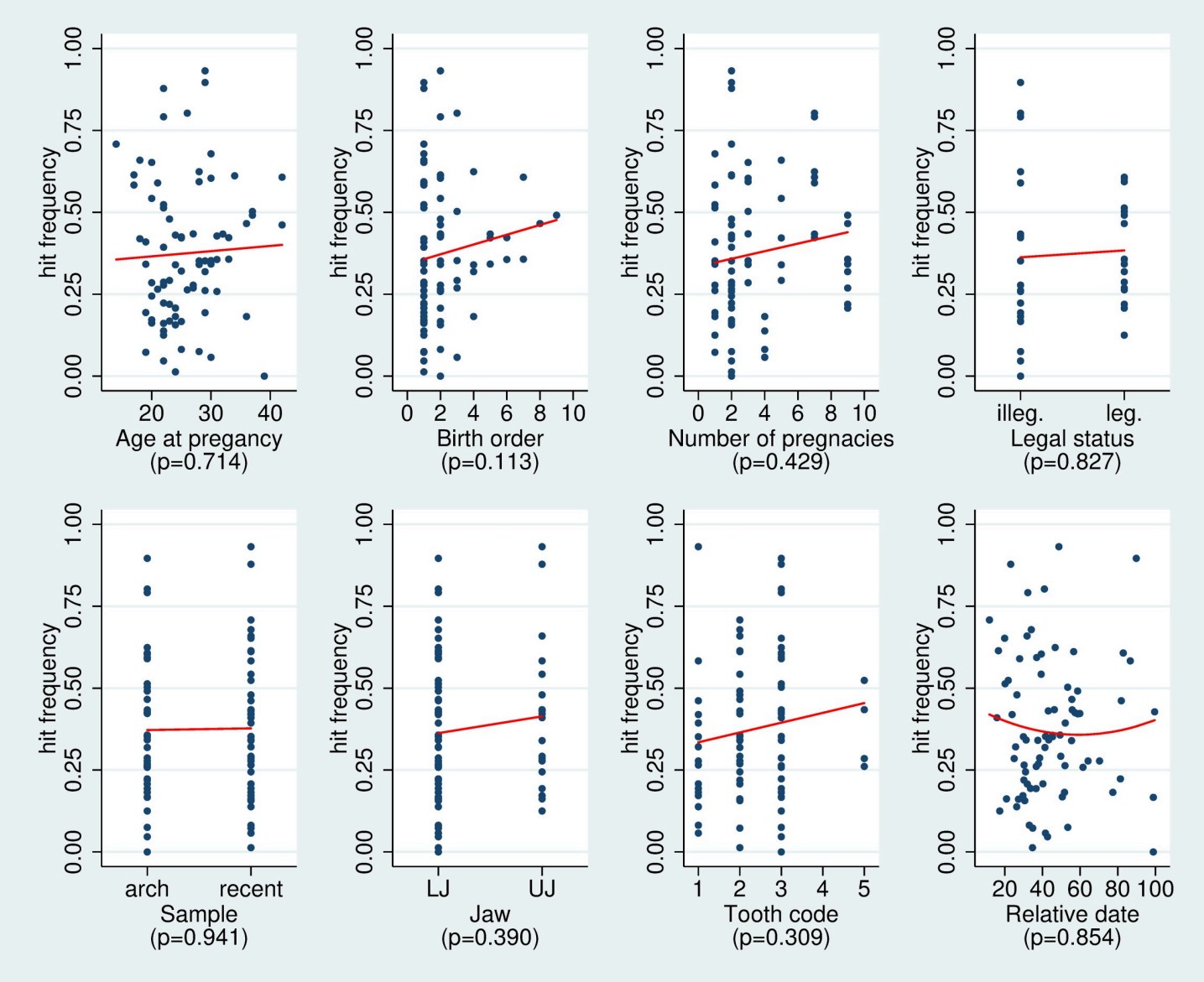

**Fig 5. Scatter plots of the hit frequency of each pregnancy vs characteristics of the pregnancy, of the mother or of the tooth for each of the three analyses performed.** The red line indicates the trend based on fitting a regression model. For the relative date, a quadratic model was used. The p-values refer to testing the null hypothesis of no association based on a regression model taking the potential clustering within individuals and across the three analyses into account.

close to a pregnancy, and often more irregularities than expected by chance. Actually, depending on the variants of signals derived from the observed irregularities–in particular by widening the age range of signals–we may find in up to 100% of all pregnancies an irregularity close to the pregnancy. Such widening was inevitable considering that the age at pregnancy based on the number of ILs provides only an approximate age estimation. However, just looking at the sensitivity may be misleading because a concordance in time between a pregnancy and an irregularity may happen by chance. When taking this into account, we observed sensitivities in the magnitude of 20 percentage points above chance level.

Methodological issues may also contribute to the moderate results. As pointed out in the systematic investigation of the measurement properties of the identification procedure [24],

**Table 5. Association of signal intensity with the number of pregnancies.**

|  | β | 95%-confidence interval | p-value |
|---|---|---|---|
| Number of S1-signals | 0.040 | [-0.23,0.31] | 0.773 |
| Number of S2-signals | 0.193 | [-0.09,0.48] | 0.193 |
| Number of S3-signals | 0.024 | [-0.18,0.23] | 0.820 |
| Number of S4-signals | -0.024 | [-0.11,0.06] | 0.556 |
| Number of S5-signals | 0.020 | [-0.12,0.16] | 0.777 |
| Number of S6-signals | 0.096 | [-0.13,0.32] | 0.400 |
| Number of S7-signals | 0.022 | [-0.07,0.12] | 0.642 |
| Number of S8-signals | 0.132 | [-0.02,0.28] | 0.095 |
| Number of S9-signals | 0.091 | [-0.16,0.35] | 0.486 |
| Number of S10-signals | 0.055 | [-0.10,0.21] | 0.484 |
| Number of S11-signals | 0.008 | [-0.14,0.16] | 0.919 |
| Number of S12-signals | 0.058 | [-0.09,0.20] | 0.441 |

$\beta$ refers to the estimated increase in the number of signals per pregnancy. The p-value refers to a test of no association.

irregular appearances and in particular local peaks were reproducible across sections only to a limited degree. This could diminish the chance of detecting pregnancies when summarizing signals across sections. However, our methodology to detect signals across section included a structured and standardized approach with visual judgement and should have managed to identify many of the relevant signals. Nevertheless, due to the moderate reproducibility of irregularities it is possible that some irregular ILs were not present in the assessed sections and therefore not captured. A further challenge is the age at signals allocation considering that the number of ILs can only provide an approximate age estimation and may limit the chance of seeing an overlap with the age at pregnancy. Therefore, we evaluated the signals also without assigning an age to signals and compared them with the number of pregnancies at the tooth level. Here we observe a positive association of the number of pregnancies with the number of signals suggesting that each fifth pregnancy is associated with a signal. This is in line with the above mentioned results. Hence, it is very likely that not all pregnancies leave traces in the tooth cementum in a way that is detectable by our structured approach, or other stimuli may affect the cementogenesis in a way that prevent or mask the irregularities caused by pregnancies.

Our investigation suggests first recommendations with respect to derive signals from irregularities of ILs in the cementum. The most promising signals tend to involve Appearances (S8, S9) than local deviations in the width (S10, S11, S12). Hence, Appearances seem to be the more promising markers to define signals to catch pregnancies.

With respect to the placements of ILs over the time span from eruption until extraction/ death no clear recommendations can be provided e.g. to count from the start to the end or vice versa. We can only recommend performing some widening of the age range of one IL beyond a single year to catch pregnancies.

## The influence of characteristics of a pregnancy on its detection

We found a slight trend of an increased hit frequency (frequency of coverage of pregnancies by signals) with increasing age and birth order. The first result is in line with previous reported results [6], the latter is in contrast. In our study we observed a tendency to a better detection of pregnancies closer to the eruption or extraction/death date. This may underline that ILs are

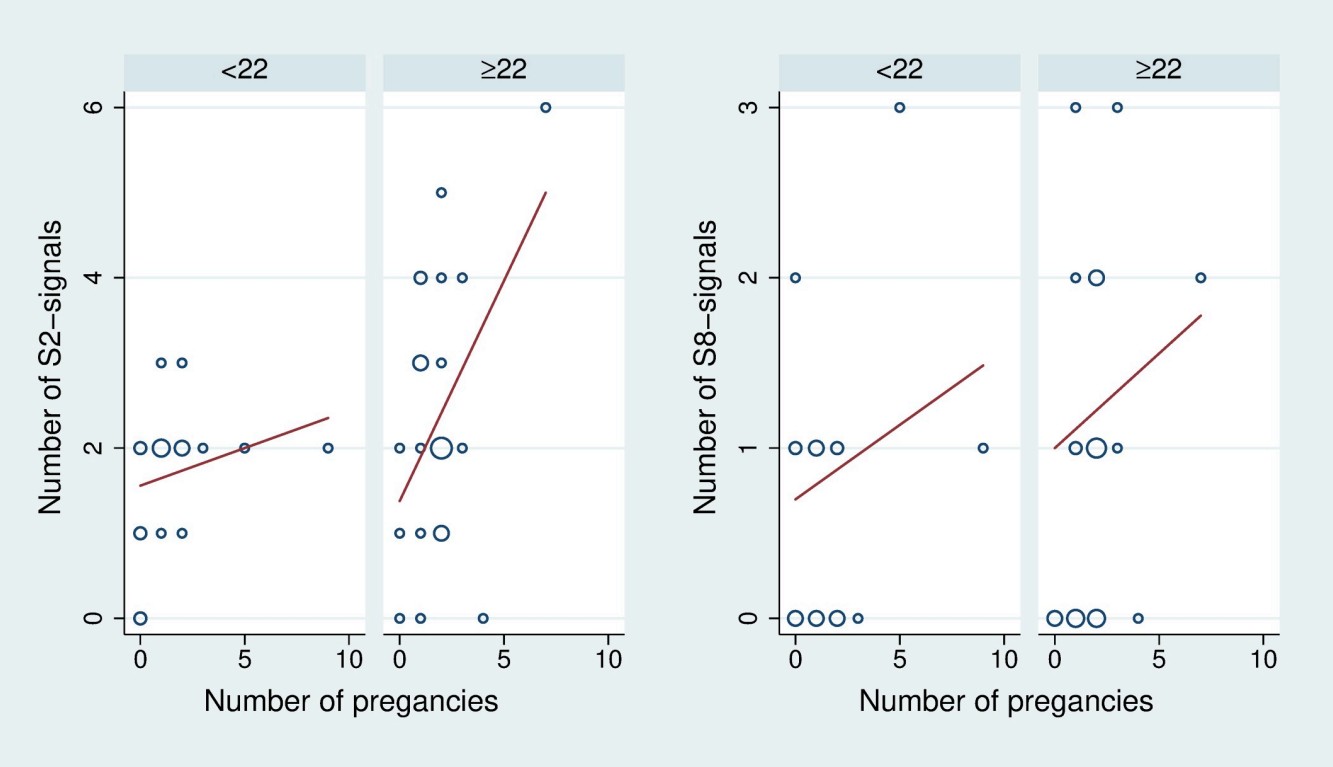

**Fig 6. Scatterplots of the number of signals by the number of pregnancies.** The scatterplots are stratified by the number of ILs per tooth in order to take into account that the number of signals and the number of pregnancies are both associated with the number of ILs.

easier to assign a correct age to if they appear close to the two available benchmarks: The (estimated) age at eruption and the age at extraction/death.

## Limitations of the study

The number of ILs of a tooth is not a perfect surrogate for the number of years between eruption and extraction/death. We observed in our population distinct differences, which were too large to be explained by the use of an estimate for the age at tooth eruption. This makes it necessary to work with rather wide age ranges to place an IL into the time span from eruption to extraction/death. Consequently, most derived signals tend to declare a large proportion of the life span of a woman as a pregnancy. In particular, if we want to reach a reasonable sensitivity above 50%, typically more than 30% of the lifespan between the 13[th] and the 45[th] birthday of a woman need to be declared as pregnancy. Actually, the women were pregnant only during 4.7% of this time span. Consequently, for all derived signals this limits their utility in practice. However, it should be emphasized that working with wide age ranges to take the uncertainty of age estimation based on the number of identified ILs into account was only necessary in order to validate signals. This is not necessary when we just want to detect pregnancies and to know their rough position within the life span.

In identifying the pregnancies in each woman, we had to rely on self-reports and historical sources. Both may be incomplete, in particular with respect to abortions and stillbirths. However, in our analyses we focused on whether and how many signals we can find close to a pregnancy. This strategy remains valid even if we do not have knowledge of all pregnancies. On the other side, it relies on correct reporting of dates of the pregnancies. However, as this seems to

be a rather safe assumption in our context, we made use of self-reports in the recent sample and detailed historical sources in the archaeological sample (S1 Text).

The strength of this study is that we assessed how many pregnancies left traces in the tooth cementum as identifiable irregular ILs based on an in detail described methodology reproducible by other researchers allowing them to assess the methodology and address potential deficiencies

## Implications for using irregularities of incremental lines in palaeodemographic research

Considering that our approach was only able to identify 20% of the pregnancies above chance level and produced often many more signals than documented pregnancies it cannot yet be applied as a routine tool to identify pregnancies. It has to be clarified, whether the moderate ability to detect pregnancies observed in this study can be improved by a methodology which assesses irregularities in the tooth cementum in a further enhanced way, or whether only a fraction of all pregnancies–perhaps the most stressful ones–can be identified. The difficulty is that cementogenesis and factors affecting it are not well understood and therefore it is difficult to distinguish between two scenarios:1) Not all pregnancies leave markers; 2) The applied methodology is not able to identify all pregnancy markers. With respect to the latter point, we observed a moderate reproducibility of irregularities across sections in our previous study [24]. Hence, a first step would be to develop a protocol for the assessment of irregularities diminishing the variation across sections, before further investigations can be started. In our study we also observed that the age estimation based on the number of identified ILs deviated from the actual age substantially in some subjects, hence an improvement of this methodology may also contribute to a better translation of irregularities to a specific age of the individual.

However, our results showed that some pregnancies are identifiable in the tooth cementum. This finding is in line with previous evidence, which successfully identified pregnancies in humans [5–7] and reproduction in animals [1–4] in the tooth cementum, and also in a recently published systematic investigation [8]. In that investigation for six out of eight parturitions a signal could be identified in the cementum within a time span of +/- 5 years, i.e. a sensitivity of 75% was reported. Unfortunately, the authors did not perform a systematic search for signals over all ILs, and hence we cannot relate this number to the signal prevalence as done in our investigation.

In the archaeological sample we saw that pregnancies with the highest detection frequency were illegitimate births. Unmarried pregnant women had to report themselves to the authorities, as an illegitimate pregnancy was a criminal offence. Accordingly, the women were subjected to an interrogation and the reports were recorded. Psychological stress during pregnancy such as anxiety can lead to adverse outcomes in mothers [28]. We observed in our previous research [24] that in the archaeological sample, including a higher proportion of women from the lower class [29] compared to the recent group, more irregular ILs in younger years were present and here, too, the question arises of whether this can be explained by demanding life conditions, also considering the low quality of life in European cities in the 19$^{th}$ century [30]. There is similar evidence that demanding conditions lead to irregular ILs [23, 31] or impact the tooth cementum [22]. The layered structure of cementum and the underlying causes have been assessed in many species but have not yet been fully explained [32]; more research is needed to explain such stress related impact on the cementogenesis.

In addition, for many variants of defining source signals more signals than pregnancies could be identified. The term "stress marker" itself already suggests that such markers can be related to other types of stress than pregnancy. In the case analysis, we saw, for example a very

good overlap of a signal identified in the adolescence with the death of the mother or the father of two individuals. This shows the need to look into other events that could influence the cementogenesis. For example, a relation to menopause, systemic illnesses and incarceration has been observed [8]. We plan corresponding investigations in the archaeological part of our population by considering famines, such as the great hunger crisis of 1816/17, the influence of diseases, the death of the husband and the death of the parents as stressful events [15]. In the best case, we can also identify characteristics of the signals which can help to distinguish between the different types of events causing the signals.

Our results show that signals based on irregular ILs can be an indication of pregnancies and adds to the evidence of the potential of tooth cementum as a resource to identify various conditions and as a tool for paleopathologists to support the reconstruction of the life and health of past populations and assess differences in terms of stress level between populations. However, our investigation also suggests that even with a structured approach, the manual visual detection of irregularities remains challenging and potentially insufficient. It might be a promising alternative to use semi-automatic approaches in order to achieve an objective assessment with limited resources. Such approaches have been suggested for counting ILs (e.g. [33]), but their extension to detecting irregularities is still lacking.

## Conclusions

We could confirm based on a standardized and reproducible method that pregnancies leave visible signals in the tooth cementum. In particular changes in the appearance of the ILs showing the potential of tooth cementum to record life history markers. There is some evidence that not all pregnancies lead to irregular ILs in appearance and width detectable by visual inspection even if a structured approach is used. Presumably only the most stressful pregnancies leave such a signal. Further research is necessary to define potential markers in the cementum and to explore their relation to pregnancies and other potentially stressful events in the lifetime of individuals. The well documented Basel-Spitalfriedhof collection used in this study is ideal for such research.

## Supporting information

**S1 Text. Source research for the reference skeleton series „Basel-Spitalfriedhof".**
(PDF)

**S2 Text. Process to manually identify potential signals.**
(PDF)

**S3 Text. Manual assessment of identified signals and overlap with documented pregnancies: Cases.**
(PDF)

**S4 Text. Performance of the different variants in the construction steps of the derived signals.**
(PDF)

**S5 Text. Descriptions of datasets.**
(PDF)

**S1 Data. Final output of signal identification.**
(CSV)

**S2 Data. Information on each tooth / individual.**
(CSV)

**S3 Data. Information on each pregnancy.**
(CSV)

**S4 Data. Translation table for assumed date of eruption.**
(CSV)

**S5 Data. Signals according to the different definitions and their time range.**
(CSV)

**S6 Data. Sensitivity and prevalence of each signal definition (also under chance level).**
(CSV)

**S1 Results. Outputs of the regression analyses shown in Table 5.**
(PDF)

## Acknowledgments

We thank the team from the department of Biological Anthropology, University of Freiburg i. Br. for the collection of the recent teeth, tooth section preparation, providing the images of the archaeological teeth sections, and the use their equipment. We also thank Professor Dr. Jörg Schibler from the Department of Integrative Prehistory and Archaeological Science of the University of Basel for the research support. Additionally, we would like to thank the Citizen Science Basel (CSB) (https://www.ipna.unibas.ch/bbs), for their volunteer work on basic research concerning historical sources, and especially the team of genealogists who have collected and validated the background information of the patients: Marie-Louise Gamma, Diana Gysin, Odette Haas, Ludwig Huber and Marina Zulauf (coordinating genealogist). We also thank Verena Fiebig-Ebneter, in charge of the patient-database (PDB) "Bürgerspital 1840–1870"; the Natural History Museum of Basel: Basil Thüring, Dr. Loic Costeur and Tandra Fairbanks-Freund for the English revision; the Staatsarchiv Basel-Stadt: Esther Baur, Hermann Wichers and team and the Archäologische Bodenforschung Basel-Stadt: Guido Lassau, Norbert Spichtig and team. A thank goes also to Esriel Mottus who supported the graphic layout of the figures.

Gabriela Mani-Caplazi was an employee of Takeda Pharmaceuticals AG, Zurich, Switzerland during the preparation of this manuscript.

## Author Contributions

**Conceptualization:** Gabriela Mani-Caplazi, Gerhard Hotz.

**Data curation:** Gabriela Mani-Caplazi, Ursula Wittwer-Backofen, Gerhard Hotz.

**Formal analysis:** Gabriela Mani-Caplazi, Werner Vach.

**Investigation:** Gabriela Mani-Caplazi.

**Methodology:** Gabriela Mani-Caplazi, Werner Vach.

**Project administration:** Gabriela Mani-Caplazi, Ursula Wittwer-Backofen, Gerhard Hotz.

**Resources:** Gerhard Hotz.

**Supervision:** Werner Vach, Ursula Wittwer-Backofen, Gerhard Hotz.

**Validation:** Gerhard Hotz.

**Visualization:** Gabriela Mani-Caplazi.

**Writing – original draft:** Gabriela Mani-Caplazi.

**Writing – review & editing:** Gabriela Mani-Caplazi, Werner Vach, Ursula Wittwer-Backofen, Gerhard Hotz.

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
