## [Decision Letter · Decision Letter 0]

27 Apr 2022

PONE-D-22-09813The concordance of signals based on irregular incremental lines in the human tooth cementum with documented pregnancies: Results from a systematic approachPLOS ONE

Dear Dr. Vach,

Thank you for submitting your manuscript to PLOS ONE. After careful consideration, we feel that it has merit but does not fully meet PLOS ONE’s publication criteria as it currently stands. Therefore, we invite you to submit a revised version of the manuscript that addresses the points raised during the review process.

We look forward to receiving your revised manuscript.

Kind regards,

Gwen Robbins Schug

Academic Editor

PLOS ONE

Journal Requirements:

a) Did participants provide their written or verbal informed consent to participate in this study?

3. We noted in your submission details that a portion of your manuscript may have been presented or published elsewhere. 

"The manuscript is part of the first author's PhD thesis, which have been accepted by the Unversity of Basel. However, the thesis is not published. It will be made available to the public in April 2023."

Additional Editor Comments:

Thank you for submitting your manuscript to PLOS ONE. It is my pleasure to inform you that it has been accepted for publication pending you address a few minor revisions suggested by the Reviewers. I really look forward to handling the revision and to seeing this paper published if you are able to address these comments satisfactorily. I will evaluate the revisions. The paper will not need further review.

Reviewers' comments:

Reviewer's Responses to Questions

**Comments to the Author**

1. Is the manuscript technically sound, and do the data support the conclusions?

Reviewer #1: Yes

Reviewer #2: Partly

2. Has the statistical analysis been performed appropriately and rigorously? 

Reviewer #1: Yes

Reviewer #2: Yes

3. Have the authors made all data underlying the findings in their manuscript fully available?

Reviewer #1: Yes

Reviewer #2: Yes

4. Is the manuscript presented in an intelligible fashion and written in standard English?

Reviewer #1: Yes

Reviewer #2: Yes

5. Review Comments to the Author

Reviewer #1: A sound and painstaking study based on unique material, dedicated to check "reverse" hypothesis: if pregnancies leave detectable traces on dental cementum incremental lines, maybe it is possible to determine past pregnancies from incremental lines? Authors show that notwithstanding all efforts relation absent but poor and insufficient for use in e.g.paleodemography (or forensic identification).

Minor comments: more details on protocol how teeth sections were performed and tools used for visualisation (now only reference given - please check the publication date, it is missing). I would suggest also more technical details from the main text (especially Methods section) move to Supplements

Reviewer #2: Investigation of cementum variation to identify pregnancy is a promising are of interest, yet a very complicated one considering all the potential confounding factors. This particular team of researchers has been at the forefront of cementum studies for over 20 years, and their contribution is always welcome. However, the methodology developed in their previous papers to define IL, and this follow up seem rather cumbersome, maybe unnecessarily so. What they define as a standardized, reproducible method, is as far as I am concerned, not efficient enough, and simply too cumbersome to be of any really use, especially in paleodemography. However, I welcome the rigorous approach and systematic review of limits of the study that does contribute to move research forward. However, I am a bit surprised that they have disregarded entirely a whole body of evidence (Newham and colleagues 2021; 2022; Newham and Naji 2022) that has already proven to be potentially more informative (but not necessarily applicable at a wider scale) regarding the influence of pregnancies on cementum variation. This is disappointing considering that the one of the authors (UWB) has contributed to one edited volume covering the topic (Naji, Rendu and Gourichon 2022).

Finally, my main criticism would be that the combination of the uncertainty linked to the authors original IL definition, combined with the potential biological variability of cementum growth will probably never result in a precise identification method dedicated to pregnancies. Potentially, any major growth alteration variable might have a confounding effect on a visual inspection of a AEFC cementum semi-thin section. Using a male control group would have been a powerful argument to justify (or discard) the original IL method in terms of picking out signals.

Nevertheless, as stated previously, any contribution at that level that moves the debate forward is welcome.

6. PLOS authors have the option to publish the peer review history of their article (what does this mean?). If published, this will include your full peer review and any attached files.

Reviewer #1: No

Reviewer #2: No

---

## [Author Response · Author response to Decision Letter 0]

30 Jun 2022

All points raised by the reviewer and editor are addressed in the uploaded response letter.

---

## [Editor Report · Decision Letter 1]

1 Aug 2022

The concordance of signals based on irregular incremental lines in the human tooth cementum with documented pregnancies: Results from a systematic approach

PONE-D-22-09813R1

Dear Dr. Vach,

We’re pleased to inform you that your manuscript has been judged scientifically suitable for publication and will be formally accepted for publication once it meets all outstanding technical requirements.

Kind regards,

Gwen Robbins Schug

Academic Editor

PLOS ONE

Additional Editor Comments (optional):

I am pleased to accept your paper and look forward to seeing it published in PLOS ONE.
---

## [Editor Report · Acceptance letter]

1 Sep 2022

PONE-D-22-09813R1 

The concordance of signals based on irregular incremental lines in the human tooth cementum with documented pregnancies: Results from a systematic approach 

Dear Dr. Vach:

I'm pleased to inform you that your manuscript has been deemed suitable for publication in PLOS ONE. Congratulations! Your manuscript is now with our production department. 

Kind regards, 

on behalf of

Dr. Gwen Robbins Schug 

Academic Editor

PLOS ONE